Plastome data reveal multiple geographic origins of Quercus Group Ilex

Simeone Marco Cosimo 1 mcsimeone@unitus.it
Grimm Guido W. 2
Papini Alessio 3
Vessella Federico 1
Cardoni Simone 1
Tordoni Enrico 4
Piredda Roberta 5
Franc Alain 6 7
Denk Thomas 8
1 Department of Agricultural and Forestry Science (DAFNE), Università degli Studi della Tuscia , Viterbo , Italy
2 Department of Palaeontology, University of Wien , Wien , Austria
3 Dipartimento di Biologia, Università degli studi di Firenze , Firenze , Italy
4 Department of Life Science, Università degli studi di Trieste , Trieste , Italy
5 Stazione Zoologica Anton Dohrn , Napoli , Italy
6 INRA, UMR BIOGECO-1202 , Cestas , France
7 UMR BIOGECO-1202, Université Bordeaux , Talence , France
8 Department of Palaeobiology, Swedish Museum of Natural History , Stockholm , Sweden
Reimer James
Electronic publication date: 2016 Apr 21
Publication date: 2016
Volume: 4
Electronic Location ID: e1897
Received 2015 Dec 28; Accepted 2016 Mar 15
Copyright: ©2016 Simeone et al.
Copyright year: 2016
Copyright holder: Simeone et al.
License: This is an open access article distributed under the terms of the Creative Commons Attribution License, which permits unrestricted use, distribution, reproduction and adaptation in any medium and for any purpose provided that it is properly attributed. For attribution, the original author(s), title, publication source (PeerJ) and either DOI or URL of the article must be cited.
License URL: https://creativecommons.org/licenses/by/4.0/

Keywords: Fagaceae, Mediterranean, Ancient introgression, Incomplete lineage sorting, Decoupled phylogenies

Funding: Swedish Research Council Austrian Science Fund M-1751-B16 This project was funded by a Swedish Research Council (VR) grant to TD. GWG is financed by the Austrian Science Fund (FWF), grant M-1751-B16. The funders had no role in study design, data collection and analysis, decision to publish, or preparation of the manuscript.

==============================
Nucleotide sequences from the plastome are currently the main source for assessing taxonomic and phylogenetic relationships in flowering plants and their historical biogeography at all hierarchical levels. One major exception is the large and economically important genus Quercus (oaks). Whereas differentiation patterns of the nuclear genome are in agreement with morphology and the fossil record, diversity patterns in the plastome are at odds with established taxonomic and phylogenetic relationships. However, the extent and evolutionary implications of this incongruence has yet to be fully uncovered. The DNA sequence divergence of four Euro-Mediterranean Group Ilex oak species (Quercus ilex L., Q. coccifera L., Q. aucheri Jaub. & Spach., Q. alnifolia Poech.) was explored at three chloroplast markers (rbcL, trnK/matK, trnH-psbA). Phylogenetic relationships were reconstructed including worldwide members of additional 55 species representing all Quercus subgeneric groups. Family and order sequence data were harvested from gene banks to better frame the observed divergence in larger taxonomic contexts. We found a strong geographic sorting in the focal group and the genus in general that is entirely decoupled from species boundaries. High plastid divergence in members of Quercus Group Ilex, including haplotypes shared with related, but long isolated oak lineages, point towards multiple geographic origins of this group of oaks. The results suggest that incomplete lineage sorting and repeated phases of asymmetrical introgression among ancestral lineages of Group Ilex and two other main Groups of Eurasian oaks (Cyclobalanopsis and Cerris) caused this complex pattern. Comparison with the current phylogenetic synthesis also suggests an initial high- versus mid-latitude biogeographic split within Quercus. High plastome plasticity of Group Ilex reflects geographic area disruptions, possibly linked with high tectonic activity of past and modern distribution ranges, that did not leave imprints in the nuclear genome of modern species and infrageneric lineages.

Introduction

Quercus L. (oaks) is among the most ecologically diverse and economically important extratropical tree genera in the northern hemisphere (Kubitzki, 1993). Quercus is the largest genus in the order Fagales, comprising ca. 400–500 species (Govaerts & Frodin, 1998). Species diversity is highest in the Americas (Groups Quercus, Lobatae and Protobalanus; Flora of North America Editorial Committee, 1997) and Southeast Asia and southern China (Group Cyclobalanopsis; Flora of China Editorial Committee, 1999). In contrast, a relatively lower number of species occurs in western Eurasia and the Mediterranean (Groups Ilex and Cerris; Kubitzki, 1993; Menitsky, 2005). The six major infrageneric lineages of Quercus occur from the tropics to the high mountains of the temperate zone and to the boreal continental, cold temperate regions (Denk & Grimm, 2010). The northern limit of oaks in North America and Eurasia coincides with the border of Dfb to Dfc and Dwb to Dwc climates, snow climates with warm versus cool summers (Köppen, 1936; Kottek et al., 2006; Peel, Finlayson & McMahon, 2007).

Recent molecular phylogenetic studies at and below the genus level focussed on the nucleome of oaks (Oh & Manos, 2008; Denk & Grimm, 2010; Hipp et al., 2014; Hubert et al., 2014). These studies consistently recovered two main lineages, the ‘New World Clade’ comprising the white oaks (Group Quercus), red oaks (Group Lobatae) and golden-cup oaks (Group Protobalanus), and the ‘Old World Clade’ consisting of the cycle-cup oaks (Group Cyclobalanopsis), the Ilex oaks (Group Ilex) and the Cerris oaks (Group Cerris). Evidence from nuclear markers and the fossil record suggests that the initial split in the ‘New World Clade’ was pre-Oligocene between the lineages leading to Group Lobatae and Group Protobalanus/Quercus (Bouchal et al., 2014; Hubert et al., 2014; Grímsson et al., 2015). This early radiation of the Quercus/Protobalanus lineage left its imprints in the molecular signatures of the few modern species of Group Protobalanus and two narrow endemic white oak species, Quercus pontica (north-eastern Turkey, south-western Georgia; Denk & Grimm, 2010) and Q. sadleriana (California; Hubert et al., 2014). Within the ‘Old World Clade’, the major split was established between the evergreen Groups Cyclobalanopsis and Ilex during the Eocene/Oligocene, whereas the chiefly temperate Group Cerris is suggested to have evolved (‘budded’) from a Group Ilex stock, possibly in Europe, not before the earliest Miocene (Denk & Grimm, 2009; Kmenta, 2011; Hubert et al., 2014; Velitzelos, Bouchal & Denk, 2014).

Nuclear amplicon data sets have also contributed to resolve the circumscription of these six groups and to delineate some intergroup and interspecies relationships (López de Heredia et al., 2007; Pearse & Hipp, 2009; Denk & Grimm, 2010; Hubert et al., 2014); well-resolved within-lineage relationships were recently obtained from phylogenomic data in the genetically least-diverged, but species-rich Group Quercus (Hipp et al., 2014). Nucleome-based studies, therefore, clearly indicate a strong correlation between morphology/speciation and nuclear differentiation in oaks. In contrast, oak plastid haplotypes are extensively shared between groups of related species (Whittemore & Schaal, 1991; Belahbib et al., 2001; Petit et al., 2002; Kanno et al., 2004; López de Heredia et al., 2007; Okaura et al., 2007; Neophytou et al., 2010; Gugger & Cavender-Bares, 2013). Notably, this was also observed in other genera of Fagaceae such as Fagus (Fujii et al., 2002; Lei et al., 2012; Zhang et al., 2013b) and Lithocarpus (Cannon & Manos, 2003), and other Fagales such as the northern hemispheric Carya (Juglandaceae; Zhang et al., 2013a) and the South American Nothofagus (Nothofagaceae; Acosta & Premoli, 2010; Premoli et al., 2012). Plastomes of this large group of long-lived woody plants appear to retain molecular signatures of evolutionary events that cannot be investigated when considering the nuclear DNA alone (e.g., Cavender-Bares et al., 2011; Premoli et al., 2012). As such, they can provide additional information to complement hypotheses on diversification and speciation processes. However, the extent and evolutionary implications of nuclear-plastome incongruence in Quercus have yet to be fully uncovered.

Testing the potential of DNA barcoding in western Eurasian oaks, Simeone et al. (2013) recently found puzzling diversity in the plastid haplotypes of samples belonging to Group Ilex. Several members of this oak group appeared paraphyletic with Groups Cerris and Quercus, and an underlying geographic partitioning was suggested in addition to the detected interspecific haplotype sharing. In the present study, we increased the geographic coverage and taxon sampling to explore the complex patterns of plastome evolution in Quercus Group Ilex. This species group is today confined to extra-tropical regions of Eurasia, spanning from arid Mediterranean maquis to high mountain and sub-alpine Himalayan forests and thickets, and to subtropical forests of SE Asia. Group Ilex includes some 35 evergreen, mostly sclerophyllous taxa, whose taxonomy is still controversial (see Table 1) and biogeographic history is not yet well understood (Menitsky, 2005; Denk & Grimm, 2010). In this work, we compiled plastid sequence data for 81 accessions of 20 oak taxa of Group Ilex. The main sampling effort was put into the four species currently occurring in the Mediterranean and adjacent regions in North Africa (Atlas Mountains) and northern Turkey (Black Sea region): the widespread Quercus ilex L. and Q. coccifera L., and the two East Mediterranean narrow endemics Q. aucheri Jaub. & Spach. and Q. alnifolia Poech. Data for additional 56 individuals of ca. 40 species were also produced to integrate all subgeneric Quercus groups and their worldwide geographic distribution. Additionally, Fagales data sets were harvested from gene banks to allow interpretation of the observed divergence in the plastid markers within a larger taxonomic frame. Our objectives were: (1) to assess the extent of plastome diversity in the Euro-Mediterranean focal group; (2) to outline key phylogeographic patterns within Quercus Group Ilex; (3) to establish major evolutionary steps for the differentiation of the ‘Old World Clade’.

Table 1 Species list.

Species included in Quercus Group Ilex according to Denk & Grimm, (2010); nomenclature followed Govaerts & Frodin (1998); species investigated in the present study are bolded. Taxonomic remarks and species distributions according to Govaerts & Frodin (1998).

Species	Taxonomic remarks	Distribution	
Q. acrodonta Seemen	Includes Q. handeliana A. Camusa/b	C, E and S China	
Q. alnifolia Poech		Cyprus	
Q. aquifolioides Rehder & E.H.Wilson	Includes Q. semecarpifolia subsp. glabraa	Tibet, C and SW China to Myanmar	
Q. aucheri Jaub. & Spach		SW Anatolia	
Q. baloot Griff.		Pakistan, Afghanistan	
Q. baronii Skan	Numerous morpho-ecological traits in common with members of Group Cerrisa	NC and SW China	
Q. bawanglingensis C.C. Huang, Ze X. Li & F.W. Xing	Poorly known; uncertain status, related to Q. phillyreoidesb	SE China	
Q. coccifera L.	Includes Q. calliprinos Webba	Mediterranean	
Q. cocciferoides Hand.-Mazz.	Includes Q. taliensis A. Camusb	CS China	
Q. floribunda Lindl. ex A. Camus	Basionym: Q. dilatata Lindl. ex A.DC. nom. illegit.	Pakistan, Afghanistan, Nepal	
Q. dolicholepis A. Camus	Includes Q. fimbriatab	CW to SW China	
Q. engleriana Seemen		Tibet to E China, Myanmar	
Q. fimbriata Y.C. Hsu & H. Wei Jen	Included in Q. semecarpifoliaa or Q. dolicholepisb	C China	
Q. franchetii Skan		C China to N Vietnam	
Q. gilliana Rehder & E.H. Wilson	Included in Q. spinosab	Tibet, C and S China	
Q. guyavifolia H. Lév.	Included in Q. semecarpifolia subsp. glabraa; Q. pannosa or Q. aquifolioides var. rufescensb	C and S China	
Q. ilex L.		Mediterranean	
Q. kingiana Craib		C China to N Thailand	
Q. lanata Sm.	Included in Q. leucotrichophorab	Buthan to Vietnam	
Q. leucotrichophora A. Camus	Basionym: Q. incana Roxb. nom. illegit.	N Pakistan, N India, Nepal to N Vietnam	
Q. lodicosa O.E. Warb. & E.F. Warb.		SE Tibet to Myanmar	
Q. longispica (Hand.-Mazz.) A. Camus	Includes Q. semecarpifolia subsp. glabraa; Q. rehderianab	C and S China	
Q. marlipoensis Hu & Cheng	Poorly knowna; very close to Q. englerianab	C China	
Q. monimotricha Hand.-Mazz.		C and S China	
Q. oxyphylla Hand.-Mazz.	Includes Q. spathulata Seemen; included in Q. dolicholepisb	C and S China	
Q. pannosa Hand.-Mazz.	Possibly conspecific with Q. semecarpifoliaa	C and S China	
Q. phillyreoides A. Gray	Includes Q. utilisa	C China to Japan	
Q. pseudosemecarpifolia A. Camus	Includes Q. semecarpifolia subsp. glabraa, Q. rehderianab	Tibet to CS China	
Q. rehderiana Hand.-Mazz.	Included in Q. semecarpifolia subsp. glabraa; includes Q. longispica and Q. pseudosemecarpifoliab	Tibet to C and S China	
Q. semecarpifolia Sm.	Includes: Q. fimbriata, Q. gujavifolia, Q. aquifolioides, Q. rehderiana, Q. longispica, Q. pseudosemecarpifoliaa	Afghanistan to Myanmar	
Q. senescens Hand.-Mazz.		E Himalaya, Tibet, C and S China	
Q. setulosa Hickel & A. Camus		C China to Vietnam	
Q. spinosa David		NC and SW China to Taiwan	
Q. tarokoensis Hayata		E Taiwan	
Q. utilis Hu & Cheng	Included in Q. phillyreoides subsp. fokiensisa	C China	
Notes.

a Menitsky (2005).

b Flora of China Editorial Committee (1999).

Material and Methods

Plant material, DNA amplification and sequencing

Our analysis included 59 individuals of the four Mediterranean Quercus Group Ilex species (File S1) covering their entire range in North Africa and western Eurasia. Additionally, 22 individuals of 16 Asian species of Group Ilex were analysed. The final dataset also included all species of the western North American Group Protobalanus (five species, 10 individuals), 16 species of Group Quercus (20 individuals, from North America and Eurasia), five species of the East Asian Group Cyclobalanopsis (11 individuals), seven species of the American Group Lobatae (eight individuals), and six species of Group Cerris (seven individuals). The outgroup set was represented by one sample each of the monotypic genera Notholithocarpus and Chrysolepis (western North America) and one species each of Castanea and Castanopsis (NCBI GenBank accessions HQ336406 (complete plastid genome of C. mollissima), JN044213, JF941179, FJ185053). Based on their genetic (plastid) signatures these genera are the closest relatives of Quercus within the Fagaceae (Manos, Cannon & Oh, 2008). For voucher information and accession numbers see File S1. The molecular analyses included three plastid DNA regions: a part of the rbcL gene, the trnH-psbA intergenic spacer and a portion of the trnK/matK region (3′ intron and partial gene). These markers were chosen based on the variability displayed in previous analyses (e.g., Manos, Zhou & Cannon, 2001; Okaura et al., 2007; Simeone et al., 2013) and the high number of their sequences available on GenBank. Primer sequences for the three regions were obtained from Kress & Erickson (2007), Shaw et al. (2005) and Piredda et al. (2011), respectively. DNA extractions and PCR protocols were the same as in Piredda et al. (2011). Sequencing of both DNA strands was performedat Macrogen (http://www.macrogen.com), using the forward and reverse PCR primers,; electropherograms were edited with CHROMAS 2.3 (http://www.technelysium.com.au) and checked visually.

Assessment of overall diversity in Quercus and Fagaceae

The diversity of the investigated regions was evaluated with Mega 5.2 (Tamura et al., 2011) and DnaSP5.1 (Librado & Rozas, 2009). For comparisons of divergence patterns across all Fagales, available data in gene banks were processed using gbk2fas (Göker et al., 2009); multiple sequence alignments were done with mafft v.7 (Katoh & Standley, 2013) using default settings and checked by eye to remove inconsistencies and erroneous sequences (taxa and sequence numbers are given in the Online Supporting Archive; www.palaeogrimm.org/data/Smn15_OSA.zip). To minimise the effect of alignment gaps, and since we were primarily interested in assessing intra- and intergeneric divergence, alignments included only subsets of the Fagales: (1) Nothofagaceae (data covering all four genera); (2) Fagaceae (10 genera including Quercus); (3) Betulaceae-Ticodendron-Casuarinaceae (11 genera); (4) Juglandaceae (9 genera); (5) Myricaceae (4 genera). Pairwise distance matrices (uncorrected p-distance, K2P, HKY, GTR+Γ) for each marker were calculated with PAUP* 4.0 (Swofford, 2002). Minimum intra-specific and minimum/maximum inter-specific distances (calculated with g2cef; Göker & Grimm, 2008) within and between genera, subgenera in the case of Fagus, and infrageneric groups in case of Quercus, are listed in File S2.

Phylogenetic analyses

Multiple sequence alignments for the focal group were obtained with ClustalW 1.81 (Thompson, Higgins & Gibson, 1994) and checked by eye. The matrices were concatenated with the Python programme combinex2_0.py (Python v. 2.6.4; Biopython 1.57).

Maximum likelihood trees were inferred with GARLI (Zwickl, 2006; run on the CIPRES portal, http://www.phylo.org/sub_sections/portal/) using four data partitions (rbcL and matK codons, trnK intron and trnH-psbA spacer). MrModeltest 2.0 (Nylander, 2004) and the Akaike Information Criterion (AIC; Akaike, 1974) were used to decide on the best-fitting substitution model for each partition.

MrModeltest 2.0 results were also used for setting up Bayesian inference, performed with MrBayes 3.4b4 (Ronquist & Huelsenbeck, 2003; Ronquist et al., 2012). RAxML v. 7.0.4 (Stamatakis, Hoover & Rougemont, 2008) was used for calculating maximum likelihood bootstrap support (1,000 replicates). Trees were edited with Figtree 1.3.1 (Rambaut, 2014) and Mesquite v. 2.75 (Maddison & Maddison, 2011). Median-joining (MJ) haplotype networks were inferred with Network 4.6.1.1 (http://www.fluxus-engineering.com/) for each gene region (rbcL, trnK/matK, trnH-psbA), treating gaps either as missing or 5th state. The MJ algorithm was invoked with default parameters (equal weight of transversion/transition), in order to handle large datasets and multistate characters.

Primary data, analyses, results and Files S1–S3 are provided as online supporting archives at www.palaeogrimm.org/data/Smn15_OSA.zip.

Table 2 Diversity values and models of DNA evolution of the fragments used for the analyses in 59 Quercus species (137 individuals) and 4 outgroup taxa.

Markers	AL	P	Nhap	Hd	S	PICs	θw	π	ME	
rbcL	743	0.00–0.008	28	0.846 ± 0.027	26	18	0.0063	0.0027	HKY+ I	
trnH-psbA	634	0.00–0.035	37 (84)	0.944 ± 0.008	38	23	0.0159	0.009	GTR+ G	
trnK/matK	705	0.00–0.022	49 (51)	0.952 ± 0.008	59	31	0.0156	0.0064	n.d.	
trnK (intron)	401	n.d.	32 (34)	0.821 ± 0.028	36	16	0.0169	0.0048	GTR+ G	
matK (codons)	304	n.d.	25	0.925 ± 0.008	23	15	0.0146	0.0086	HKY	
rbcL+trnK/matK+trnH-psbA	2,082	0.00–0.014	74 (110)	0.978 ± 0.005	122	72	0.0119	0.0056	Combined	
rbcL +matK	1,047	n.d.	49	0.965 ± 0.006	49	33	0.0085	0.0044	n.d.	
trnH-psbA +trnK	1,035	n.d.	57 (103)	0.954 ± 0.008	69	34	0.0161	0.0067	n.d.	
trnH-psbA +trnK-matK	1339	n.d.	65 (110)	0.970 ± 0.006	92	49	0.0155	0.0072	n.d.	
Notes.

AL Aligned length (bp)

P uncorrected p-distance (min.–max.)

N hap Number of identified haplotypes, brackets: with gaps considered

Hd Haplotype diversity

S Number of polymorphic sites

θ Nucleotide polymorphism

π Nucleotide diversity

PIC Number of Parsimony Informative Characters

ME Model of evolution

Results

Levels of intra- and interspecies plastome divergence in Quercus

The entire dataset included 423 plastid DNA sequences (141 samples, three markers each). The sequence quality was high for all the marker regions: 100% of unambiguous full-length electropherograms with 100% overlap between complementary sequences were recovered across all taxa. Table 2 shows that trnH-psbA was the most variable marker region (a 34-bp inversion occurring in approximately 50% of the samples was replaced with its reverse-complementary sequence and a binary character was inserted to keep record of it). As expected, the least variable region was rbcL. No indels were found in the rbcL and matK coding regions. The combined cpDNA dataset (trnH-psbA, trnK/matK, rbcL) resulted in an alignment of 2082 characters (sites), of which 122 were variable (thereof 72 parsimony-informative; gaps not considered). The combined regions had a nucleotide diversity of 0.006 and included 74 different haplotypes of which 50 were unique (restricted to a single accession). As a result, the overall haplotype diversity was high (Hd = 0.978 ± 0.005). With gaps considered, the number of haplotypes increased to 110, of which 89 were unique (Hd = 0.994).

In general, the infrageneric divergence calculated in Quercus is comparable to that found in other genera of the Fagaceae and Betulaceae, and higher than in Juglandaceae (Table 3). All three gene regions allow distinguishing the generic affinity of an oak individual; the same haplotype may be shared by several or many oak species (usually within the same infrageneric group; Table 3), but not with other genera of the Fagaceae.

Table 3 Divergence patterns in other Fagaceae and Fagales.

Divergence patterns in Quercus compared to other Fagaceae and Fagales based inter-species pair wise uncorrected p-distances of sequences retrieved from GenBank and produced in this study.

			Intrageneric divergence	Mean intergeneric divergence at family levelc	
			rbcL	matK	trnH-psbA	rbcL	matK	trnH-psbA	
Taxon	Nt	Ns	Min.	Max.	Min.	Max.	Min.	Max.	Min.	Max.	Min.	Max.	Min.	Max.	
Quercus	87/87/86	219/255/382	0.000	0.010	0.000	0.023	0.000	0.042	0.003	0.011	0.006	0.015	0.006	0.042	
Fagus	8/9/6	33/30/19	0.000	0.014	0.000	0.007	0.000	0.042	0.024	0.036	0.091	0.098	0.120	0.147	
Other Fagaceae	37/21/28	102/32/86	0.000	0.020	0.000	0.021	0.000	0.020	0.003	0.013	0.000	0.019	0.011	0.032	
Nothofagaceaea	23/b /14	35/b /53	0.000	0.027	–b	–b	0.000	0.017	0.012	0.023	–b	–b	0.017	0.041	
Betulaceae	55/19/77	131/34/247	0.000	0.011	0.000	0.006	0.000	0.069	0.006	0.024	0.014	0.033	0.011	0.079	
Juglandaceae	18/b /21	23/b/28	0.000	0.005	–b	–b	0.000	0.007	0.000	0.021	–b	–b	0.006	0.034	
Notes.

Nt number of taxa

Ns number of sequences

a Values for rbcL may be over-estimated (data usually older than 15 years; sequences show features characteristic for sequencing and editing artifacts).

b Insufficient data.

c Values for Quercus and other Fagaceae not including Fagus (see Fagus for max. inter-generic divergence in Fagaceae).

At the infrageneric level in Quercus, minimal inter-species distances can be zero for all three markers and within all infrageneric groups. Notably, maximal inter-species distances within infrageneric groups of Quercus can reach or even exceed the level of inter-generic differentiation in Fagaceae (e.g., between Notholithocarpus, Lithocarpus, Castanopsis, Castanea, Chrysolepis), Juglandaceae and Myricaceae. The maximum intra-specific distance found in Mediterranean individuals of Quercus Group Ilex equals the maximum inter-specific divergence found within this group (Table 3; for additional data see File S2).

Phylogenetic placement of Mediterranean Quercus Group Ilex plastid haplotypes

Individuals of the Mediterranean species of Quercus Group Ilex cluster in three well supported clades (Fig. 1). The first clade (‘Euro-Med’) accommodates most accessions of Q. ilex and Q. coccifera. In the second clade (‘Cerris-Ilex’), accessions of Q. ilex, Q. coccifera, and one of the five samples of Q. aucheri group together with all representatives of Quercus Group Cerris and two Himalayan-East Asian species of Group Ilex. Sister to this clade are the three representatives of the single Japanese species of Group Ilex (Q. phillyraeoides). In the third clade (West Asia-Himalaya-East Asia; ‘WAHEA’) the remaining specimens of Q. aucheri form a subclade along with the Cypriote endemic Q. alnifolia, and several Eastern Mediterranean Q. coccifera. The second, more divergent and poorly supported subclade comprises two western Himalayan species (Q. baloot, Q. floribunda), two individuals of Himalayan-East Asian species of Quercus Group Ilex, and one Central China accession of a Cyclobalanopsis member (Q. oxyodon) sympatric with many group Ilex oaks, including Q. semecarpifolia, Q. leucotrichophora and Q. floribunda (Menitsky, 2005). In contrast to Group Ilex, all other infrageneric groups show relatively high chlorotypic coherence, usually forming clades or grouping within the same subtree. The actual root of the tree is obscured; representatives of Castanea, Castanopsis, and Notholithocarpus/Chrysolepis that could be used as putative outgroups are placed in different subtrees.

Figure 1 ML tree of the investigated oak accessions.

ML tree of plastid accessions; tentatively rooted with the Notholithocarpus-Chrysolepis subtree. Stars indicate subtrees comprising accessions of Mediterranean members of Quercus Group Ilex. Colouration refers to the taxonomic affiliations and main clades of specimens. Number at branches indicate non-parametric bootstrap support under maximum likelihood using two different implementations and posterior probabilities calculated using Bayesian inference.

Evolutionary significance of plastid haplotypes in western Mediterranean oaks of Quercus Group Ilex

The MJ network for the plastid region with the highest overall variability (trnH-psbA, only length-homogenous parts considered; Fig. 2; MJ networks for the other gene regions can be found in File S3 and in the online archive www.palaeogrimm.org/data/Smn15_OSA.zip) highlights the evolutionary significance of the three main haplotypes, ‘Euro-Med’, ‘Cerris-Ilex’, and ‘WAHEA’. Three main clusters differ by a minimum of two conserved mutations: (1) Group Quercus, Protobalanus and Lobatae (‘New World Oaks’); (2) individuals with ‘Euro-Med’ haplotypes; (3) individuals with ‘Cerris-Ilex’ and ‘WAHEA’ haplotypes, representatives of Group Cerris and East Asian species of Group Ilex and Group Cyclobalanopsis (‘Old World Oaks’). In general, haplotypes (File S3 includes MJ-networks for the other three regions, rbcL gene, matK gene, 3’ trnK intron) found in the western Eurasian members of Group Ilex represent unique or ancestral variants. Unique haplotypes of Group Cerris are directly derived from the Group Ilex or shared ‘Cerris-Ilex’ haplotypes. Haplotypes of Group Cyclobalanopsis are identical to or can be derived from East Asian members of Group Ilex. The graphs further highlight a close relationship of haplotypes of Chrysolepis and Notholithocarpus with those of the ‘New World’ oaks; the haplotypes of Castanea and Castanopsis can be derived from the ‘Old World’ oaks basic type.

Figure 2 Haplotype network based on the trnH-psbA spacer.

Haplotype network based on length-conserved portions of the trnH-psbA spacer. Colouration refers to the taxonomic affiliation of specimens.

Figures 1 and 2 clearly illustrate that differentiation in the plastid sequences of Quercus (and related Fagaceae) is independent from the formation of the modern clades or, at least, from their genetic homogenization (lineage sorting).

Phylogeographic structure in Quercus Group Ilex

Haplotypes forming the ‘Euro-Med’, ‘Cerris-Ilex’ and ‘WAHEA’ lineages are geographically sorted but shared among species. The phylogenetically isolated ‘Euro-Med’ haplotypes are encountered in the western Mediterranean populations of Q. ilex and Q. coccifera (North Africa, Iberia, Southern France, Italy), along the Adriatic coast and into Central Greece (Fig. 3). Also included here are isolated populations of Q. ilex from Crete and the southern Black Sea coast. ‘Cerris-Ilex’ and ‘WAHEA’ haplotypes are confined to the eastern Mediterranean region. ‘Cerris-Ilex’ haplotypes are found in the Aegean region (Q. ilex, Q. coccifera and Q. aucheri individuals) and replaced by ‘WAHEA’ haplotypes (Q. coccifera, Q. aucheri, Q. alnifolia) in south-western Turkey and extending to the east (Levante region; Fig. 3). The ‘Cerris-Ilex’ type is also found in the Q. coccifera individual from northern Turkey, representing the north-easternmost population of this species.

Figure 3 Plastid haplotype variation in Mediterranean members of Quercus Group Ilex.

Geographic pattern of plastid haplotype variation in Mediterranean members of Quercus Group Ilex. (A) Map showing the taxonomic identity of sampled specimens. (B) Map showing the plastid haplotypes of sampled specimens.

Discussion

All currently available molecular data on Fagaceae show a deep incongruence between nuclear and plastid data. Nuclear phylogenies unambiguously point towards an inclusive common origin of all oaks, i.e., a monophyletic (s. str.) genus Quercus (Oh & Manos, 2008; Denk & Grimm, 2010; Hubert et al., 2014). At the same time plastid data repeatedly failed to resolve all oaks as one clade (Manos, Cannon & Oh, 2008; this study). Instead, a split emerges (with varying support) between the North American Notholithocarpus and a North American/northern temperate clade of oaks, the ‘New World Oaks’, and the Eurasian Castanea, Castanopsis and oak lineages, the ‘Old World oak’ clade. If we accept the monophyly of the genus Quercus, which is further supported by morphology and evidence from the fossil record, haplotypes of Castanea/Castanopsis and Notholithocarpus that group with the ‘New World’ and ‘Old World’ oaks, respectively, can only be the result of incomplete lineage sorting during the formation of the modern genera and/or ancestral gene flow between early diverged lineages. In addition, the plastid gene pool of the earliest oaks must have shown a genetic gradient that reflected to some extent a biogeographic pattern. Although it is impossible to pinpoint the place of origin of oaks, it is clear that paleoclimatic and paleogeographic conditions during the Eocene facilitated a rapid spread over the Holarctic region, allowing them to pick up and propagate geographic signatures inherited from their common ancestors with Notholithocarpus, Castanea and Castanopsis.

Major trends of plastome differentiation

Overall low genetic intra- and intertaxonomic (intrageneric lineages, genera) distances suggest low evolutionary rates for the chloroplast genomes of Fagales at the examined loci. However, the data coverage is far from sufficient for most genera and families to precisely assess the potential variation within the plastome of this plant group. In Fagaceae, a comparison with the (genetically) more diverse Nothofagaceae and Betulaceae families shows that haplotype variation at the trnH-psbA locus can be sufficiently high to allow phylogeographic and systematic inferences (see Premoli et al., 2012; Grimm & Renner, 2013). We observed similar levels of variation for haplotypes of intrageneric lineages of Quercus at this marker. Furthermore, a geographic pattern is evident for the most widely sampled groups. Groups Ilex, Lobatae and Quercus were the most variable groups, whereas Group Cerris exhibited the lowest variation rates among its species. Interclade differentiation among all Quercus groups equalled or exceeded that found in the four outgroup genera (Castanea, Castanopsis, Notholithocarpus and Chrysolepis). As a consequence, the outgroup taxa appear scattered across the tree, rather than being culled in a distinct subtree, rendering the plastome of Quercus ‘non-monophyletic’. Outgroup selection as a potential source of topological ambiguity has previously been pointed out by Hubert et al. (2014) who analyzed 108 oak taxa at eight nuclear markers. Ambiguous relationships within Fagales independently of the strength of the obtained phylogenetic signal were also suggested in a recent study based on plastid DNA, fossil and reproductive syndromes which resolved the majority of inter-generic relationships in each family except for in the Quercoideae group making Castanopsis and Quercus “non-monophyletic” (Xiang et al., 2014).

Figure 4 Map of chloroplast evolution in oaks.

Mapping of chloroplast evolution in oaks (using the same rooting scenario as in Fig. 1) on current evolutionary synopsis (based on nuclear sequence data, morphology, and the fossil record; modified after Grímsson et al. (2015, Fig. 16). Colouring of the plastid lineages refers to branches/subclades in Fig. 1: bluish, common (ancestral) and ‘New World’ oak/castanoids plastid haplotype lineages; green, lineages of the unique ‘Euro-Med’ plastid haplotype found only in Mediterranean members of Group Ilex; reddish, lineages of ‘Old World’ oaks and Eurasian castanoids. Note that members of Group Ilex keep plastid haplotypes of five different evolutionary sources/systematic affinities. Abbreviations: C, Cretaceous; Pa, Paleocene; E, Eocene; O, Oligocene; M, Miocene; Pl, Plio-/Pleistocene.

How polyphyletic is Quercus Group Ilex?

Figure 4 highlights the incongruence of the plastid genealogy tree with the current understanding of the evolution of Fagaceae and oaks based on molecular sequence data from non-coding nuclear gene regions (Manos, Zhou & Cannon, 2001; Denk & Grimm, 2010), a recent time-calibrated nuclear phylogeny of oaks (Hubert et al., 2014), and the fossil record of modern lineages as documented by pollen investigated under the scanning-electron microscope (Grímsson et al., 2015; see also Denk & Grimm, 2009). Extensive sampling is more likely to reveal polyphyly (Wiens & Servedio, 2000), but caution is needed when trees are interpreted regarding single (monophyly, paraphyly) or multiple (polyphyly) origins of species. Indeed, polyphyly as expressed in phylogenetic trees, ‘non-monophyly’, is more likely due to homoplasy between distant branches in a tree. Additional reasons for topological polyphyly is that we work with gene trees, and not species trees: in this case, topological polyphyly (‘non-monophyly’) may be due to ancestral polymorphisms or introgression causing incongruences between the trees. However, distinguishing the effects of these mechanisms may be very difficult in the absence of nuclear markers and (palaeo-)geography as complementary information (Funk & Omland, 2003). As a general rule, deep putative polyphyly or paraphyly (lineages resolved as a “basal” grade) hints at retained ancestral polymorphism, i.e., a lineage sorting phenomenon, while recently introgressed haplotypes may assume a highly derived position in a gene tree. At the same time, incomplete lineage sorting is not predicted to promote the geographic proximity of interspecifically shared haplotypes that may be seen under local introgression (Hare & Avise, 1998; Masta et al., 2002).

A strictly polyphyletic origin of Quercus Group Ilex or its Mediterranean members is unlikely. Nuclear data covering the entire range of Q. ilex and Q. coccifera in the Mediterranean region unambiguously resolved the two species as close, but mutually monophyletic sister taxa (Denk & Grimm, 2010). The ‘non-monophyly’ of Group Ilex plastomes seen in the tree (Fig. 1), including haplotypes shared with Group Cerris or closely related to Group Cyclobalanopsis therefore reflects either incomplete lineage sorting or introgression or both. In the absence of nucleome data for all here included individuals, it is impossible to infer to which degree introgression and incomplete lineage sorting contributed to the plastid gene pool of the Mediterranean species of Group Ilex. Nevertheless, the most straightforward explanation for the observed scenario would be a combined effect: asymmetrical introgression of ancestral haplotypes resulting in local genetic clusters decoupled from taxonomic boundaries, in which plastome accessions of species or species complexes may form grades or multiple clades in phylogenetic trees, thus appearing para- or polyphyletic (e.g., Rieseberg & Soltis, 1991; Whittemore & Schaal, 1991).

As modelled by Excoffier, Foll & Petit (2009), interspecific interactions during historical range fluctuations can profoundly affect the observed phylogeographic patterns, and manifest as paraphyly or reticulation (polyphyly in a broad sense). In fact, most range expansions do not occur in completely uninhabited areas, and interbreeding between local and an expanding (invasive) species with subsequent asymmetrical introgression can develop also in absence of selection, e.g., when one species is dominant and most abundant (Lepais et al., 2009). Plastid haplotypes referring to the original (‘lost’) species are indeed likely to persist over long evolutionary periods, and may still be found in the invading species. Noteworthy, environmental changes and disturbance of local communities have been shown to increase hybridisation rates (Lagache et al., 2013), hence, the potential for widespread, imbalanced introgression. In Group Ilex oaks, the interspecific capture of plastids among sexually incompletely isolated species likely occurred on the geological timescale, concealing the species relationships at various stages in the history of the genus. In a comprehensive study of the genus Ilex (Manen et al., 2010), the high incongruence between a taxonomically compatible nuclear gene tree and a geographically structured plastid tree was explained with extensive extinctions between the Cretaceous and Miocene and multiple hybridization and introgression events between distantly related lineages. This has been documented also for Platanus (Grimm & Denk, 2010) and more recently suggested for the evergreen white oaks of Quercus subsection Virentes (Eaton et al., 2015). Similar ancient lateral transfers have been also inferred to explain the paraphyly of the maternally inherited mtDNA of Picea species (Bouillé, Senneville & Bousquet, 2011) and Pinus species (Tsutsui et al., 2009).

Decoupling of plastid signatures and taxonomy in oaks

Speciation processes in Quercus do not immediately leave imprints in the plastome (e.g., Neophytou et al., 2010; Cavender-Bares et al., 2011) as also well documented for Nothofagus (Acosta & Premoli, 2010; Premoli et al., 2012). Low mutation rates and long generation times can contribute to slow evolutionary rates and incomplete lineage sorting of organellar genomes (Cavender-Bares et al., in press; Besnard, Rubio de Casas & Vargars, 2007). In addition, reiterated extinction and re-colonisation involving bottlenecks, genetic drift, and founder effects may cause random fixation of haplotypes, increasing the probability for retaining ancestral traits. Oaks in general, and especially the Mediterranean taxa, are also characterised by a marked resprouting ability in response to environmental disturbances (Barbero et al., 1990). This could have contributed to clonally preserve and transmit ancestral plastid lineages (maternally inherited) during multiple unfavourable conditions since the origin of the Mediterranean region (Blondel & Aronson, 1999). At the same time, large population sizes and long distance pollen dispersal might have contributed to homogenise the nuclear genomes in local populations of a species but not their organelle genomes. Topographic barriers hindering seed dispersal and balancing selection on the organelle genomes could have preserved ancestral polymorphisms within the species’ gene pools, also (Funk & Omland, 2003). However, although there is some evidence for balancing selection on the mtDNA in plants (Städler & Delph, 2002) it has not yet been suggested as an explanation of intraspecific plastid heterogeneity.

Additionally, Fagaceae lineages are susceptible to hybridisation and introgression (Arnold, 2006). This may lead to the formation of morphologically unambiguous individuals of a species with plastid signatures of another (Whittemore & Schaal, 1991; Petit et al., 2004). There is strong evidence for local introgression in oak communities with morphologically distinct species in the case of European white oaks (Group Quercus; Q. robur, Q. petraea, Q. pyrenaica, Q. pubescens, Q. frainetto; Curtu, Gailing & Finkeldey, 2007; Valbuena-Carabaña et al., 2007; Lepais et al., 2009), as well as in members of Quercus subsection Virentes, a subgroup of Group Quercus, in North America (Cavender-Bares et al., in press), and across a wide range of Group Lobatae (Dodd & Afzal-Rafii, 2004; Peñaloza-Ramírez et al., 2010; Moran, Willis & Clark, 2012; Valencia-Cuevas et al., 2015). In our focal group, hybrids and different levels of genetic introgression among morphologically pure individuals were molecularly documented via genetic assignment analysis in Q. ilex/Q. coccifera (Ortego & Bonal, 2010) and, to a lesser extent, in Q. coccifera/Q. alnifolia (Neophytou et al., 2011). Also, the potential for inter-group hybridisation was experimentally demonstrated for Q. ilex and Q. robur (Group Quercus; Schnitzler et al., 2004), and natural introgression in Q. ilex and Q. suber (Group Cerris) was identified in southern France (Mir et al., 2009) and the Iberian Peninsula (Burgarella et al., 2009). Therefore, it is possible that ancient hybridization and introgression, favoured by the well-known sexual promiscuity between closely related taxa and their ability to disperse pollen over long distances, obscure the interpretation of the evolutionary origin of an oak species or entire lineage.

In the Mediterranean, the profound geological and ecological changes during the Neogene (Blondel & Aronson, 1999) likely caused extinction, re-colonisation, range fragmentation and hybridisation linked to secondary contact, especially when species were still young and reproductive barriers likely weaker than today. Taken together, incomplete sorting of ancestral traits and introgression of haplotypes thus appear highly likely mechanisms to decrease inter-species plastid differentiation while at the same time increasing intra-species variation.

Temporal and spatial framework of plastome evolution

The three distinct plastid haplotypes observed in modern Mediterranean members of Quercus Group Ilex may reflect three radiation phases (range extensions), followed by range disruptions and isolation of plastome lineages within the ‘Old World Clade’ of Quercus. Considering the high diversity of haplotypes in Group Ilex as compared to other major oak lineages (or other genera in the Fagales; see Table 3; File S2) it can be hypothesized that the geographical disruptions in the plastome of the ancestors of Group Ilex and interacting lineages predate the manifestation of modern taxa (species and infrageneric groups; Fig. 1). Haplotypes shared between members of Group Ilex and its sister lineages Group Cerris and Group Cyclobalanopsis may indicate the same geographic origin or may be the result of secondary contact and asymmetrical introgression.

Evolutionary hypotheses concerning the unique ‘Euro-Med’ haplotype

Independent from the actual position of the plastid root (note the scattered placement of putative outgroups in Fig. 1), the divergence of the ‘Euro-Med’ haplotype must have coincided with the initial differentiation in Quercus (Fig. 1). Oaks had achieved a wide northern hemispheric range by the Eocene. Unequivocal fossils are known from high latitudes (North America, Greenland, North Europe; Crepet & Nixon, 1989; Manchester, 1994; Grímsson et al., 2015) and mid-latitudes (Central Europe, South East Asia; Kvaček & Walther, 1989; Hofmann, 2010). All major lineages of oaks were established by the end of the Eocene, ca. 35 Ma, as evidenced by the fossil record and molecular dating using eight nuclear gene regions (Bouchal et al., 2014: Fig. 14; Hubert et al., 2014; Grímsson et al., 2015). During this time, one fraction of oaks, represented by the ‘Euro-Med’ plastids, must have been geographically and reproductively isolated which would have caused a major split in the plastid gene pool (Fig. 1). Today, the ‘Euro-Med’ haplotype within Quercus Group Ilex is the only one exclusively shared by just two, but widespread Mediterranean species of Group Ilex, Q. ilex and Q. coccifera. This haplotype is markedly distinct from haplotypes in other oaks or Fagaceae (Fig. 1). Two evolutionary hypotheses could explain the establishment of this unique haplotype in Q. ilex-Q. coccifera (Fig. 5): (i) The ‘Euro-Med’ haplotype is the remnant of an extinct oak lineage that was intrograded (invaded) and consumed by members of Group Ilex (Fig. 5B); under this scenario Group Ilex would have migrated into Europe at some point prior to the Miocene where it came into contact with this extinct oak lineage. (ii) The ‘Euro-Med’ haplotype represents an original plastome of Group Ilex; under this scenario, the first split within the modern ‘Old World clade’ would have been between western members of Group Ilex and an eastern cluster comprising precursors, members of Group Ilex and Group Cyclobalanopsis (Fig. 5C). At the moment we can only speculate which hypothesis applies.

Figure 5 Origin of the ‘Euro-Med’ haplotype.

Eocene set-up and the origin of the ‘Euro-Med’ haplotype. (A) Unequivocal fossil record of oaks in the Eocene mapped on a palaeotopographic map highlighting a primary split into a high-latitude and mid-latitude lineage that likely correspond to the deep phylogenetic split seen in nuclear and plastid sequence data of modern oaks between the ‘New World Clade’ (Groups Protobalanus, Quercus and Lobatae) and the ‘Old World Clade’ (Groups Cyclobalanopsis, Ilex, Cerris). (B–C) Scenarios that can explain the occurrence of the unique ‘Euro-Med’ haplotype in westernmost members of Quercus Group Ilex. (B) The ‘Euro-Med’ haplotype belonged to an extinct oak lineage geographically/biologically separated from both the ancestors of the New World and Old World Clade. Westward expansion of Himalayan members of Group Ilex and subsequent large-scale introgression/hybridisation homogenised the western members of Group Ilex and the extinct oak lineage, retaining and evolving the original haplotype in the Mediterranean region. (C) The ‘Euro-Med’ haplotype reflects geographic fragmentation within the Paleogene range of the Old World Clade that was overprinted to some degree after later radiation phases of Group Ilex. Palaeotopographic map base used with permission from Ron Blakey, © Colorado Plateau Geosystems.

During the Eocene and Oligocene a number of extinct genera of Fagaceae (e.g., Eotrigonobalanus, Trigonobalanopsis) and extinct lineages of Quercus were present in the northern hemisphere (Kohlman-Adamska & Ziembínska-Tworzydło, 2000; Kohlman-Adamska & Ziembínska-Tworzydło, 2001; Stuchlik, Ziembínska-Tworzydło & Kohlman-Adamska, 2007; Denk, Grímsson & Zetter, 2012; Grímsson et al., 2015). It is possible that the plastid of one of these lineages was captured by members of Quercus Group Ilex when expanding westwards into the modern area of the Euro-Med haplotype. This would fit with the first hypothesis.

Scenario (ii) would be in agreement with geographic fragmentation and subsequent incomplete lineage sorting across the former Paratethyan and current Himalayan arcs. During its evolution, Group Ilex must have been continuously affected by range disruptions caused by tectonic activity south of the Paratethys linked to the collision of Africa and the Indian subcontinent with Eurasia (Fig. 6); progressive rarefaction of the original haplotypes and the occurrence of (repeated) invasion and introgression events that left imprints in the plastome even within the same species is highly likely in such a topographic setting. Nevertheless, under scenario (ii) the diversity and distinctness of the ‘Euro-Med’ haplotype would require an initial geographic separation between the westernmost (European) and other members of Group Ilex, predating the formation of the modern lineages and surviving later (Miocene) migration waves. Therefore, scenario (ii) appears less probable than scenario (i).

Figure 6 Tectonic activity during the Eocene.

Tectonic activity during the Eocene and past and modern distribution of the New World (white) and Old World (yellow) groups within Quercus. Black lines indicate major subduction zones, red lines major orogenies. Note that the high latitude lineage of oaks (Quercus Group Lobatae, Group Quercus, Group Protobalanus) evolved in tectonically stable regions, whereas the low latitude lineage (Quercus Group Ilex, Group Cyclobalanopsis, Group Cerris) evolved in tectonically unstable regions. Uppercase and lowercase letters refer to extant and extinct distribution areas of major oak lineages: P,p, Group Protobalanus; Q,q, Group Quercus; L,l, Group Lobatae; I,i, Group Ilex; C,c, Group Cerris; Y,y, Group Cyclobalanopsis. Palaeotopographic map base used with permission from Ron Blakey, © Colorado Plateau Geosystems.

Evolutionary significance of the ‘Cerris-Ilex’ haplotype and the origin of Group Cerris

The ‘Cerris-Ilex’ haplotype is shared between all species of Quercus Group Cerris (western Eurasian and East Asian), East Mediterranean (Aegean) individuals and two East Asian species of Group Ilex. This is in agreement with Denk & Grimm (2010) who suggested that Quercus Group Cerris evolved from Group Ilex by budding (a hypothesis further confirmed by the 8-nuclear gene data set used by Hubert et al. (2014)), and the low support for a Group Ilex clade in an all-Fagaceae (excluding Fagus) tree based on over 1000 nuclear ITS sequences (Denk & Grimm, 2010). Hubert et al. (2014) inferred a Miocene age for this budding event, which corresponds to the earliest unequivocal fossil of Quercus Group Cerris (Kmenta, 2011) and is younger than the earliest definite fossil record of Quercus Group Ilex in Europe (early Oligocene, Cospuden; (Denk, Grímsson & Zetter, 2012)). Also, dispersed pollen from the Paleogene Changchang Formation, Hainan (Hofmann, 2010), resembles both Quercus Group Ilex and Group Cyclobalanopsis; the age of this formation is considered late early to early late Eocene (Lei et al., 1992). The haplotype most closely related to the ‘Cerris-Ilex’ haplotype is encountered in the widespread East Asian Q. phillyraeoides, the only species of Group Ilex extending to Japan (in contrast, the East Asian members of Group Cerris have a much wider range in north-eastern Asia; Menitsky, 2005). Regarding its phylogenetic position, the emergence of the ‘Cerris-Ilex’ haplotype appears to have been linked with a major taxonomic sorting event in Eurasian Fagaceae, resulting in distinct haplotypes restricted to genera and intrageneric groups of Quercus (Fig. 1). Based on the palaeobotanical record, these lineages (Castanopsis, Castanea, Quercus Group Ilex, Quercus Group Cyclobalanopsis) were well established at least by the Eocene (Table 4 and Fig. 5; Grímsson et al., 2015); a deep divergence is reflected by their distinctly different nuclear genomes (Oh & Manos, 2008; Denk & Grimm, 2010; Hubert et al., 2014). Again, two evolutionary scenarios could explain the occurrence of the ‘Cerris-Ilex’ haplotype in Aegean individuals of Q. ilex and Q. coccifera and the westernmost Q. aucheri: (i) Group Cerris evolved in western Eurasia/Himalaya from an (extinct) subtropical to temperate sublineage of Group Ilex, which left its imprint in the Aegean members of Group Ilex, and Q. spinosa, Q. engleriana and Q. phillyraeoides. (ii) Group Cerris shares a common ancestry with the north-east Asian Q. phillyraeoides of Group Ilex. Under the latter scenario, the budding event of the group would have taken place in north-eastern Asia, from where it migrated into western Eurasia and the Aegean region. In relatively recent times, Group Cerris came into contact with the Mediterranean members of Group Ilex and was locally introgressed (e.g., Burgarella et al., 2009; Mir et al., 2009). However, it is difficult to explain why Q. ilex-coccifera should only intrograde into populations of Cerris oaks at a large scale in the Aegean region. Therefore, scenario (i) appears to be more plausible. The fossil record of Group Cerris, in particular the Central and East Asian one, could potentially shed further light on this hypothesis. Unfortunately, it is currently not very well understood.

Table 4 Eocene fossil record of Quercus.

Locality, site, age	Reference	Taxon, organ	Affinity	
Clarno Fm., Oregon, western U.S.A.; ∼48 Ma	Manchester, 1994	“Quercus” paleocarpa Manchester; cupules and acorns	Group Cyclobalanopsis (? Lithocarpus)	
Axel-Heiberg Island, Canadian Arctic; ∼45 Ma	McIntyre, 1991; McIver & Basinger, 1999	Pollen and leaves	New World Clade; Quercus Group Quercus/Lobatae	
Hareøn, western Greenland; ∼42 Ma	Grímsson et al., 2015	Quercus sp. 4–5; pollen	Extinct/ancestral type	
		Quercus sp. 6–7; pollen	New World Clade (aff. Group Protobalanus)	
		Quercus sp. 1–3; pollen	Quercus Group Quercus and/or Lobatae	
Baltic amber, northern Europe; ∼45 Ma	Crepet & Nixon, 1989; Weitschat & Wichard, 2003	Flower and in situ pollen	Quercus Group Quercus	
Königsaue, near Aschersleben, Germany; middle Eocene (48–38 Ma)	Kvaček & Walther, 1989	Quercus subhercynica H Walther & Kvaček; leaf	Quercus Group Lobatae	
Ube coal-field, southwestern Honshu, Japan; middle Eocene (48–38 Ma)	Huzioka & Takahasi, 1970	Cyclobalanopsis nathoi Huzioka & Takahashi; leaf, acorn (?), cupule (?)	Quercus, affinity unclear	
Changchang, Hainan; middle (?) Eocene (50–35 Ma)	Hofmann, 2010	Quercus pollen types 2–8, 10; pollen	Quercus, affinity unclear (extinct, Group Quercus/Lobatae?, Group Protobalanus?)	
		Quercus pollen type 1; pollen	Quercus Group Ilex	
		Quercus pollen type 9; pollen	Quercus Group Cyclobalanopsis	

Origin and evolutionary significance of the ‘WAHEA’ haplotype of Group Ilex

The West Asian–Himalayan-East Asian (WAHEA) haplotype represents Eastern Mediterranean members ofQuercus Group Ilex and is sister to a clade comprising several Asian species of Group Ilex (Himalayas to the mountains of Southeast Asia). Based on its position in the plastid tree, this haplotype seems to reflect a second radiation within the Old World Clade and allies after the isolation of the ‘Euro-Med’ lineage and prior to the radiation and subsequent sorting within the clade comprising the ‘Cerris-Ilex’ haplotypes (Fig. 1). The modern distribution of species with the WAHEA haplotype follows the Himalayan corridor (Kitamura, 1955; Güner & Denk, 2012). The Himalayan corridor is a narrow band along the southern flanks of the Himalaya with a subtropical to temperate climate (Cwa, Cwb; Peel, Finlayson & McMahon, 2007) providing a pathway for plants originating from humid temperate Cenozoic laurel and mixed broad-leaved deciduous and evergreen forests. In addition to Quercus Group Ilex (Zhou, 1992; Velitzelos, Bouchal & Denk, 2014), prominent relic taxa include species of Acer, Aesculus, Cedrus, Cotinus, Juglans, Platanus, and Rhododendron among others. The ‘WAHEA’ haplotype represents the western counterpart to the haplotype lineage found in East Asian species of Group Ilex and Cyclobalanopsis (Fig. 1). The relic Q. alnifolia, today only found in the mid-montane region of Cyprus (Mt. Troodos), would be a witness of this expansion (Menitsky, 2005).

Conclusion

Plant biogeographic studies at the genus level have commonly relied on few to many chloroplast markers and a single or very few accessions per taxon. In the case of woody angiosperms with a subtropical to temperate distribution such as e.g., Nothofagaceae (Svenson et al., 2001; Knapp et al., 2005), Rhus (Yi, Miller & Wen, 2004), Cornus (Xiang et al., 2005), Carpinus (Yoo & Wen, 2007), Castanea (Lang et al., 2007), Juglans (Aradhya et al., 2007), and Carya (Zhang et al., 2013a), this approach runs the risk of capturing but a limited aspect of the evolutionary history of the focal group. Mere combination with e.g., nuclear ITS data can be problematic, too (compare data shown here with data provided by Denk & Grimm (2010), on western Eurasian members of Group Ilex). The decoupled evolutionary signals in plastomes and the nucleome/morphology as documented for Nothofagus (Acosta & Premoli, 2010; Premoli et al., 2012) and Quercus Group Ilex (this study) suggest that the traditional placeholder sampling strategy is not ideal. Signals from few-marker/many-samples data sets are likely to be complex or even puzzling (Figs. 1–4), but at the same time provide entirely new perspectives on plant evolution worth exploring. For Quercus Group Ilex, our pilot study focussing on Mediterranean species reveals a crucial aspect of oak evolution not seen in the combined nuclear, morphological, and fossil data: large-scale introgression and/or incomplete lineage sorting among ancestral lineages of modern major groups and species. The new data corroborate hypotheses that Group Cerris evolved (‘budded’) relatively recent from Group Ilex as discussed by Denk & Grimm (2010), using over 600 ITS and over 900 5S-IGS accessions covering all western Eurasian oak species and inferred by Hubert et al. (2014), using dated phylogenies based on seven single-copy nuclear regions and the ITS region. Our preferred hypothesis is that Quercus Group Cerris evolved in western Eurasia and the Himalayas when the then chiefly subtropical low latitude Group Ilex radiated into temperate niches. Accordingly, within modern members of Group Cerris, a wide spectrum of leaf traits is found from pseudo-evergreen in Q. suber, to semi-evergreen in Q. brantii, Q. ithaburensis, Q. trojana (partly) and fully deciduous in Q. acutissima, Q. castaneifolia, Q. cerris, Q. libani and Q. variabilis. The conspicuous plastid diversity in the Mediterranean species of Group Ilex and the lineage in general (Figs. 1 and 2; Table 2) can only be interpreted as reflecting the highly complex geographical history of this group, due to growing geographic isolation between clades from the Oligocene onwards. The ‘Euro-Med’ haplotype evidences an initial phase of geographic differentiation predating the formation of modern lineages, but its origin and evolutionary significance remain enigmatic (Fig. 5).

Although decoupled from taxonomy, the plastid phylogeny provides important, independent information on the geographic differentiation of Quercus prior to the formation of modern species/species groups. So far, the major split within oaks has been between ‘New World’ and ‘Old World’ oaks (following Manos, Zhou & Cannon, 2001) because of the current distribution of the major, molecular-defined lineages of oaks. This view replaced traditional concepts (reviewed in Denk & Grimm, 2010) recognising two subgenera/genera, one in subtropical to tropical East Asia (subgenus/genus Cyclobalanopsis = Group Cyclobalanopsis) and the other ubiquitous on the northern hemisphere (subgenus/genus Quercus, including all other infrageneric groups). The plastid data presented here suggest that the early evolution of oaks instead was geographically bound to high latitude Arctic regions and to low latitude subtropical regions (Fig. 5). The high latitude lineages remained genetically homogeneous in the nucleome, but also in the plastome to some degree. Continuous circum-polar distribution could have prevented pronounced genetic drift in the high latitude lineage, which became the ‘New World Clade’, and explains low genetic differentiation in deciduous high and mid latitude white oaks until today (Denk & Grimm, 2010). At the same time, the Atlantic, the proto-Mediterranean, the Paratethys and the rise of the Himalaya isolated the Eurasian low latitude lineage, and may explain the differentiation between plastome clades, known to be mirror of geographic isolation rather than species taxonomy in case of introgression. The high diversity of southern plastomes, the low diversity of northern plastomes, and the propensity of oaks to introgress, altogether built a consistent framework for deciphering the historical biogeography of this group.

Our data should only be viewed as a first step towards a more complete understanding of the biogeography and evolution of oaks. The next step would be to map the plastid variation of Quercus Group Ilex across its entire range by sampling multiple stands of the Himalayan and East Asian species to characterise the geographic and taxonomic ranges of the various plastid lineages. Finally, because of similar strong correlation between plastome differentiation and geographic distribution at the population level and the species/genus level, similar or identical plastid haplotypes typically shared between co-occurring and often distantly related taxa, polyphyletic signals and reproductive biology, the same processes could have likely played a key role in the evolutionary history of other Fagaceae (e.g., Fagus, Castanea, Castanopsis). As well, broadening the sampling efforts of phylogenetic analyses of the plastome could help decipher the speciation history of these genera. At the same time, extended nucleome investigations will be necessary to definitely assess a clear molecular phylogeny of Fagaceae.

Supplemental Information

File S1 List of specimens investigated

Click here for additional data file.

File S2 Plastome divergence in Fagales

Intra- and intertaxonomic minimum and maximum pairwise genetic distances in Fagales based on the used plastid markers.

Click here for additional data file.

File S3 RbcL and trnK-matK haplotype networks

haplotype networks of the investigated dataset based on rbcL and trnK-matK markers.

Click here for additional data file.

Supplemental Information 1 Online Supplementary Archive: primary data and analyses

Click here for additional data file.

We thank all the Directors and curators of the Botanic Gardens and Arboreta who provided the investigated material: Anthony S. Aiello, Wolfgang Bopp, Anne Boscawen, Peter Brownless, Béatrice Chassé, Laszlo Csiba, Dirk De Meyere, Holly Forbes, Anett Krämer, Isabel Larridon, Gitte Petersen, David Scherberich, Patrick Thompson, Jef Van Meulder, Michael Wall, David Zuckerman, and all the colleagues and friends who contributed to the sampling design: Jeannine Cavender-Bares, Hanno Schaefer, Charalambos Neophytou, Martina Temunovic, Gianni Bedini, Laura Genco, Enara Otaegi Veslin.

Additional Information and Declarations

Competing Interests

Author Contributions

DNA Deposition

Data Availability

The authors declare there are no competing interests.

Marco Cosimo Simeone and Guido W. Grimm conceived and designed the experiments, analyzed the data, contributed reagents/materials/analysis tools, wrote the paper, prepared figures and/or tables, reviewed drafts of the paper.

Alessio Papini analyzed the data, contributed reagents/materials/analysis tools, wrote the paper, prepared figures and/or tables, reviewed drafts of the paper.

Federico Vessella analyzed the data, reviewed drafts of the paper.

Simone Cardoni and Enrico Tordoni performed the experiments, analyzed the data.

Roberta Piredda analyzed the data, contributed reagents/materials/analysis tools.

Alain Franc and Thomas Denk analyzed the data, wrote the paper, reviewed drafts of the paper.

The following information was supplied regarding the deposition of DNA sequences:

All sequence data generated as part of this study are available on GenBank (http://www.ncbi.nlm.nih.gov/genbank/) under accession numbers LM652969– LM653098.

The following information was supplied regarding data availability:

Primary data and analyses are provided for download in the Online Supplementary Archive (OSA) at the journal’s homepage and at www.palaeogrimm.org/data/Smn15_OSA.zip. See GuideToFile.txt included in the OSA for further information. Other relevant data are within the paper and Supplemental Information.

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
