# Peer review of "Plastome data reveal multiple geographic origins of Quercus Group Ilex"

_PeerJ, doi:10.7717/peerj.1897_

## Round 0.1 · original submission · Minor Revisions

All three reviewers were generally very positive about your manuscript, and two of them have suggested some small changes to your text. As well, the third reviewer suggests that you clarify your discussion to some degree, which I also agree with. In summary, the comments are fair and positive, and the requested revisions are not major - for these reasons, my decision is "minor revisions".

·

Basic reporting

The paper is very well written and provides a thorough discussion of the results and presents testable hypotheses (scenarios).

The authors used sequence information on three chloroplast regions to infer the geographic origin of the Mediterranean Quercus Group Ilex including 55 species from all Quercus subgeneric groups as reference. As shown in other studies, chloroplast (cp) DNA regions were not suitable to distinguish between oak species which are known for their propensity to hybridize. However, the authors show the utility of cpDNA polymorphisms for phylogeographic studies and to infer the potential geographic origin of major oak lineages. Large-scale sampling and more detailed studies, especially in the Himalayan and Chinese regions are suggested to validate proposed scenarios of divergence, migration and ancient introgression in oaks. Since cpDNA differentiation and geographic distribution were strongly correlated, similar molecular signatures might be found in other Fagaceae species.
Future analyses could focus on additional more variable cpDNA regions to infer more recent historical processes. Also, nuclear DNA analyses could provide evidence of more recent introgression between related species and asymmetrical introgression.

Experimental design

The description of the experimental design is appropriate. Potential limitations as result of limited sampling in some regions and future research to fill these gaps are appropriately discussed.
Regarding the sequencing results: I suggest to report quality scores for the sequences and mention this in the Material and Methods section.

Validity of the findings

The data are sound and the statistical analyses are solid. The result provide new insight and testable hypotheses with regard to the evolution of oaks

Additional comments

Please find additonal edits and suugestions for improvement listed below. Additional handwritten edits and comments are attached.

Abstract
Line 42: It could be mentioned that these members of distinct oak lineages do not (or rarely) hybridize at present. This would strengthen the argument of ancient asymmetrical introgression and incomplete lineage sorting (as results of low mutation rates and the absence of recombination in the chloroplast genome).


Introduction

I suggest to write section (group) names (Quercus, Lobatae etc.) in italics.

“This early radiation of the Quercus/Protobalanus lineage left its imprints in the molecular signatures of the few modern species of Group Protobalanus and two narrow endemic white oak species, Quercus pontica (north-eastern Turkey, south-western Georgia; Denk and Grimm, 2010) and Q. sadleriana (California; Hubert et al., 2014).”

Please explain which molecular signatures were left. Q. pontica is differentiated from other members of the roburoid group at ITS.

Line 105: paraphyly of part of the Ilex Group to (?)

Line: Please include primers sequences or citations for the three chloroplast regions

Line 149: I suggest to change the title reflecting the purpose of these analyses.

Line 166: Under “Statistical tools” another alignment program is mentioned (MAFFT). Please explain why CLUSTALW is used for the focus group, and not MAFFT or MUSCLE. It is not entirely clear which data analyses were done for which purpose.

Line 198: Please report the same number of significant digits.

Line 210: Please refer to a Table.

Line: 233: Please explain why the MJ network is only shown for the plastid region with the highest overall variability.

Line 247: Please check this sentence (see handwritten comments).

Line 278: I think ancient introgression could be mentioned as an explanation for this pattern.

Line 292: This sentence is not clear to me.
Line 327: It should be explained how selection and adaptation can homogenize the nuclear genome. Balancing selection can maintain polymorphisms.

Line 340: It could be explained whether introgression has been inferred from gene flow or genetic assignment analyses.

Line 399: It could be mentioned that abundance of one species can result in asymmetrical gene flow.

Line 432: Please explain more.

Line 450: “Fossil evidence and available phylogenies (discussed in the following)
lend high credibility to scenario (i) as the most plausible explanation.”

I suggest to provide this evidence in the same paragraph as it have been done in other paragraphs. I lost track here.

Line 450: Please refer to Fig. 5c here.

Line 482: introgressed
Line 486: This would support scenario i.
I suggest to move the last sentence of the paragraph to the beginning of the paragraph.

“Hence, the fossil record clearly favours a western Eurasian-Himalayan origin of
Group Cerris (scenario i).”

Overall it should be made clearer which results support scenario I and scenario ii.


Line 535: The part in brackets could be written in a separate sentence.

Line 544: “The ‘Euro-Med’ haplotype evidences an initial phase of west-east differentiation in low-latitude Eurasian oaks, the ‘Old World Clade’, probably triggered by the complex topography within its potential range essentially since the Eocene (Fig. 5).”

Evidence for scenario I has to be described earlier (see above).

Line 547: Please refer to Fig. 5b

Line 588: “Both the ancient western Eurasian clade, now extinct but evidenced by the ‘Euro-Med’ haplotype, and the originally Himalayan clade had been invaded by the late Neogene by the direct ancestors of today’s Q. coccifera and Q. ilex.”

This refer to scenario i described at line 444. Clear evidence for scenario i should have been listed earlier.

Reviewer 2 ·

Basic reporting

I consider the manuscript adequately meets all the journal standards

Experimental design

I consider the manuscript adequately meets all the journal standards

Validity of the findings

I did not detect any problem with the data obtention, analyses and inferences drawn. The paper seems fairly robust.

·

Basic reporting

This article meets the criteria for basic reporting.

Experimental design

I do not take issue with the overall design, but I do question the power of the data used in the analyses. A polyphyletic origin could result in the outgroups being scattered, but it could also be a symptom of low-power/information content in the data. I also would have liked to see a little more attention brought to the inverted region found in a large number of samples.

Validity of the findings

Overall, the arguments are well-articulated and the findings valid. However, I take issue with several minor points. The first is the assertion that nucleome and plastome should share a consistent evolutionary history. It is well-documented that they evolve independently of one another, and that the phylogenetic relationships revealed by the plastome within sections of Quercus are often more representative of contemporary geographic location than the broader evolutionary history of the species. The conclusions drawn concerning polyphyly should be dialed back a little and labelled as speculation. I would also make sure that the conclusions that are drawn address the questions specifically, and reflect the fact that much more research needs to be done to support the claims made from these data. This is especially true of the proposed work in the Himalayan/Chinese highlands.

Additional comments

A little refocusing in your discussion will go a long way here. The paper is very thoughtful and well-written, just please make it clear when you are speculating and clarify some of the specific points I made in the earlier two boxes. If you do this, I think the paper will be ready for publication.

---

## Round 0.2 · accepted · Accept

The authors have answered the reviewers' concerns well, and I agree with the opinion of the reviewers that this manuscript is now acceptable for publication. I believe this is a fine piece of work, and with excellent analyses and beautiful figures, one that will add to PeerJ's growing reputation.

Please ensure that reviewer 1's small edits are addressed either in the proof stage or earlier (if possible).

·

Basic reporting

My earlier comments have been addressed in the new version. I attach some handwritten edits. Please check that the citations in the text are formatted consistently.

Experimental design

see my earlier comments

Validity of the findings

see my earlier comments

Additional comments

see my handwritten edits

·

Basic reporting

The article conforms to all the basic reporting standards.

Experimental design

The experimental design has been edited sufficiently to reflect the standards of the journal.

Validity of the findings

The validity of the findings meet all of the requirements for the journal.

Additional comments

I think this article is now ready for publication.